# Phylogenetic Marker Selection and Protein Sequence Analysis of the ORF5 Gene Product of Grapevine Virus A

**DOI:** 10.3390/plants11091118

**Published:** 2022-04-20

**Authors:** Mina Rastgou, Vahid Roumi, Emanuela Noris, Slavica Matić, Sezai Ercisli

**Affiliations:** 1Department of Plant Protection, Faculty of Agriculture, Urmia University, Urmia 5756151818, Iran; 2Department of Plant Protection, College of Agriculture, University of Maragheh, Maragheh 5518183111, Iran; vroumi@maragheh.ac.ir; 3Institute for Sustainable Plant Protection, National Research Council of Italy (IPSP-CNR), 10135 Turin, Italy; emanuela.noris@ipsp.cnr.it (E.N.); slavica.matic@ipsp.cnr.it (S.M.); 4Department of Horticulture, Faculty of Agriculture, Ataturk University, Erzurum 25240, Turkey; sercisli@atauni.edu.tr

**Keywords:** RNA binding protein, phylogenetic marker, RF distance, protein structure, conserved region

## Abstract

Grapevine virus A (GVA), the type species of the *Vitivirus* genus, is one of the causal agents of the Kober stem grooving disease of the rugose wood complex and one of the most frequently detected viruses in grapevine. There is little information on GVA gene(s) marker useful for phylogenetic analysis. To this aim, a total of 403 leaf samples were collected from vineyards of East and West Azarbaijan provinces in the Northwestern provinces of Iran during 2014–2016 and tested by DAS-ELISA and RT-PCR using ORF5-specific primers. GVA was detected in 56 symptomatic samples, corresponding to 14% of infection, while it was not detected in asymptomatic samples. The ORF5 (p10) protein sequence of eight Iranian isolates was compared to other vitiviruses, showing that the most conserved region resides in the N-terminus, carrying an arginine-rich motif followed by a zinc-finger motif. Next, to define a robust phylogenetic marker representative of the whole genome sequence suitable for phylogenetic and evolutionary studies, phylogenetic trees based on the full genome sequences of all the available GVA isolates and on individual genomic regions were constructed and compared. ORF1, which encodes the RNA-dependent RNA polymerase, was found to be the best phylogenetic marker for GVA classification and evolution studies. These results can be used for further research on phylogenetic analyses, evolution history, epidemiology, and etiology of rugose wood complex, and to identify control measures against GVA and other vitiviruses.

## 1. Introduction

Grapevine (*Vitis vinifera* L.) is one of the oldest and most widely grown fruit crops in the world. According to available data from 2018 [1], grapevine can be infected by nearly 80 viruses belonging to different families, some with very high incidences. Viruses can negatively affect plant vigor and longevity, and the quality and yield of the grapevine products [2].

Grapevine virus A (GVA) is one of the causal agents of the Kober stem grooving (KSG) disease of the rugose wood complex (RW) [3,4,5]. The virus is the species of the genus *Vitivirus* in the subfamily *Trivirinae,* family *Betaflexiviridae* [6], and has filamentous flexuous particles of 800 nm in length and 11–12 nm in diameter [7]. Its genome consists of a positive-sense, single-stranded RNA (+ssRNA) with five open reading frames (ORFs): ORF1 encoding the RNA dependent RNA polymerase (RdRp; 194 KDa), ORF2 encoding a protein with unknown functions, ORF3 encoding the movement protein (MP; 31 KDa), ORF4 encoding the coat protein (CP; 21.5 KDa), and ORF5 generating a protein of 10 kDa (p10) [8,9].

The main symptoms of GVA infection on susceptible grapevine plants include a lower plant vigor, leaf deformation, with yellowing and reddening, rugose wood, and stem pitting. The virus can also affect the yield, reducing the quality and quantity of grapes [10]. In addition to infected propagating material, GVA may be transmitted semi-persistently by different mealybugs and scales (*Pseudococcus* spp., *Planococcus* spp., *Heliococcus* spp.) [11]. The virus can be also transmitted by grafting and mechanical inoculation on different experimental hosts [12].

In diagnostics and phylogenetic studies, the importance of using whole genome sequences was recently emphasized, but sequencing a full viral genome is time-consuming and very expensive. Moreover, for many plant viruses around the world, such as vitiviruses, whole genome sequences of many isolates within the same species are not available. For these reasons, researchers frequently use partial sequences for phylogenetic analyses of virus isolates. The best gene marker(s) for phylogenetic analysis should have a strong correlation with the full genome sequences and be sufficiently informative, when whole genome sequencing is not possible [13]. GVA isolates from different regions of the world, including South Africa, Jordan, and America, have considerable variability in their RdRp and CP gene sequences [14,15,16,17,18]. Based on sequence analysis of the complete CP gene, four groups (I, II, III, IV) were reported [19,20]. Three (I, II, III) and four major clades were revealed based on partial CP and RdRp sequences by [19] and other authors [20,21,22]. Predajna and Glasa [23] grouped GVA isolates into four phylogenetic groups based on partial nucleotide sequences of CP and ORF5 genes. Moradi et al. [10] also generated three groups (I, II, III) based on complete CP gene sequences. Despite these reports, no conclusive analysis of the full genome and individual ORF levels are available for this virus. Previous studies in Iran have revealed the presence of GVA in the grape-producing areas of Fars, East Azarbaijan, West Azarbaijan, Kurdestan, Kohgiluyeh va Boyer-Ahmad and Zanjan provinces. These results were obtained by ELISA and RT-PCR using specific primers for ORF1(RdRp) and ORF4(CP), followed by complete or partial sequencing of these genes [10,24,25,26,27]; until now, no previous analyses of the ORF5 (p10) sequence of Iranian isolates have been reported. The p10 protein has great importance for GVA, acting as an RNA-binding protein, containing a basic, arginine-rich motif and a typical zinc-finger domain [9]. Moreover, it can affect the expression of symptoms in *Nicotiana benthamiana* plants and suppresses RNA silencing in its mesophyll cells [17].

In order to define a robust phylogenetic marker for GVA, we collected grapevine samples from vineyards belonging to five major local cultivars, in two main grape-producing regions of East and West Azarbaijan provinces in Northwest Iran, and compared the tree topology of the full genome and individual five genes of this virus. In addition, we analyzed the protein sequence of the ORF5 gene product (p10) of selected GVA Iranian isolates to collect information on the functional role of this protein.

## 2. Materials and Methods

### 2.1. Plant Material

Four hundred and three plant samples showing yellowing and reddening of leaves, irregular ripening of berries, and tree decline were collected during summer and autumn 2014–2016 from the major six grapevine producing areas of East and West Azarbaijan provinces (Northwest Iran), two areas accounting for nearly 70% of grape production in this country. The samples derived from five major local grape cultivars, including Fakhri, Garmiane Maragheh, Qizil Uzum and Keshmeshiye bidaneh (white berry), and Angore- siyah (red berry). The collected material, including leaves, petioles, and cane scraping from basal nodes, and midribs, was transported to the laboratory on ice and stored at 4 °C in air-proof plastic bags for a maximum of one week. Leaf petioles and cane scraping samples from two symptomless grapevines of each cultivar were also collected, serving as negative controls.

### 2.2. Double Antibody Sandwich Enzyme Linked Immunosorbent Assay (DAS-ELISA)

Plant samples including well-developed mature leaves, especially petioles and veins, were homogenized in extraction buffer (0.5 M Tris-HCl, pH 8.2, 0.8% NaCl, 2% PVP (MW 24,000), 1% PEG (MW 6000), 0.02% NaN3 and 0.05% Tween 20) at a ratio of 1:10 (*w/v*) and tested for GVA by DAS-ELISA, as described by Clark and Adams [28], using the GVA ELISA kit (Art.No.122276; Bioreba AG, Reinach, Switzerland), according to manufacturer’s instructions. IgG and conjugate were diluted 1:1000 in coating buffer and conjugate buffer, respectively. Negative and positive controls of the kit were also included. After 2 h, absorbance was measured at 405 nm (A405) using the ELISA microplate reader (Expert Plus, Hitech, Eugendorf, Austria). Each sample was analyzed in duplicate and mean readings being at least three times higher than the average value of the negative controls were considered as positive.

### 2.3. RNA Extraction and Reverse Transcription-Polymerase Chain Reaction (RT-PCR)

Total RNA was extracted from 200 mg of scraped bark tissue collected from basal nodes, petioles, and/or midribs from ELISA-positive samples, following the silica-capture method described by Foissac et al. [29], with small modifications. At the end, RNA was resuspended in 20 µL sterile water. Following assessment of RNA quality and concentration using a NanoDrop Spectrophotometer (Thermo Scientific, USA), RNA concentration was adjusted to 20 ng/µL and samples were stored at −20 °C for further studies.

cDNA was synthetized by incubating for one hour at 42 °C a mixture composed of 3 μL RNA, 2μL 10 X RT buffer (0.5 M Tris-HCl, 0.7 M KCl, 0.1 M MgCl_2_, pH 8), 1μL dNTPs (10 mmol/μL), 1μL DTT (100 mol/μL), 0.5 μL RNase inhibitor (10 mmol/μL), 0.5 μL MMuLV reverse transcriptase (200 U/μL, Thermo Fisher Scientific, Waltham, MA, USA), and 2 μL Reverse primer C7273(R) (5′-CATCGTCTGAGGTTTCTACTAT-3′) [30] (100 pmol/μL), in a final volume of 20 μL.

Five μL of cDNAs was used in a PCR reaction mix containing 10 mM of each dNTPs (Cinnagen, Iran), 1.6 mM of MgCl_2_, 1 U of *Taq* DNA polymerase (Cinnagen, Iran), 0.5 µL of primers H7038(F) (5′-AGGTCCACGTTTGCTAAG-3′) and C7273(R) (10 pmol/µL each) and 1 X PCR buffer. The cycling parameters were as follows; initial denaturation at 94 °C for 3 min, followed by 35 cycles of 94 °C for 30 sec, 56 °C for 46 sec, and 72 °C for 1 min, and a final extension step at 72 °C for 15 min. PCR was performed in a thermal cycler (Master cycler gradient, Eppendorf, Hamburg, Germany) and the expected products of 236 bp of the GVA ORF5 sequence were analyzed by electrophoresis in 1% agarose gel, following staining with FluoroVue™ Nucleic Acid Gel Stain (Smobio, Taiwan).

### 2.4. Sequencing and Phylogenetic Analysis

PCR products of eight ELISA-positive samples (from the Torkaman, Urmia, Varjoy, Nazloo, Siloo, Maragheh and Khaneh beig regions) were purified, directly sequenced by Macrogen (Seoul, Korea) on both strands and the obtained sequences compared with GVA sequences deposited in the GenBank database (http://www.ncbi.nlm.nih.gov) (accessed on 19 October 2018) using BLASTn (http://www.ncbi.nlm.nih.gov) (accessed on 19 October 2018). Nucleotide sequence similarity analyses and multiple alignments were performed using MAFFTv.7. Phylogenetic tree reconstruction was done by MrBayes 3.2.6 (HKY model) and embedded in Geneious Prime^®^ 2019.1.3 (Biomatters, Auckland, New Zealand) using default settings.

### 2.5. ORF 5 Sequence Analyses

*Pfam* (Protein families’ database) was used to analyze the partial ORF5 (p10) gene product (90 aa) of eight GVA Iranian isolates. The analysis was conducted with Weblogo 3 (http://weblogo.berkeley.edu/logo.cgi) (accessed on 21 October 2020) by submitting multiple sequence alignments (MSA) of the desired regions [31] of the p10 proteins of different GVA isolates, and also of p10 proteins of GVA isolates and the corresponding proteins of other vitiviruses, including grapevine virus B (GVB, Acc. No. CAG38877), grapevine virus K (GVK, Acc. No. YP_009389467), grapevine virus J (GVJ, Acc. No. YP_009551971), grapevine virus D (GVD Acc. No. CAA69071), grapevine virus H (GVH, Acc. No. YP_009551909), actinidia virus A (AVA, Acc. No. AET36889), and actinidia virus B (AVB, Acc. No. YP_004935362).

### 2.6. Sequence Analyses

The Linux version of MAFFTv.7 was used to align the complete genome sequences and the sequences of each individual gene of GVA isolates separately [32], with ‘Auto’ settings. Subsequently, Mesquite v.3.10 was used to generate MSA [33] manually and IQtree (Linux version) to reconstruct the phylogenetic tree, and to create MSAs of full genome sequences and of each gene sequence [34], automatically selecting the best substitution model using the ModelFinder program. To validate the phylogenetic trees, 1000 replicates of Hasegawa approximate likelihood ratio test (SH-aLRT) and Ultrafast bootstrap (UFBoot) were used. Finally, Figtree v.1.4.3 (http://tree.bio.ed.ac.uk/software/figtree/) (accessed on 25 October 2020) was used to visualize the IQtree and the reconstructed consensus trees (based on 50% majority rules) generated by MrBayes 3.2.7 [35] (at CIPRES Science Gateway [36]). Topology tree comparisons were performed by Robinson–Foulds (RF) distances using *TreeCmp* [37].

## 3. Results

### 3.1. Virus Detection

During 2014–2016, grapevine samples showing virus-like symptoms were collected from vineyards of East and West Azarbaijan provinces in Northwest Iran. Following GVA detection using DAS-ELISA, 56 out of 403 grapevine symptomatic samples (14%) from five major autochtonous grapevine cultivars tested positive for this virus (Table 1). These samples were further subjected to RT-PCR, using ORF5-specific primer pairs. The expected amplicon of 236 bp corresponding to a partial ORF5 sequence was amplified, while no amplicons were obtained from healthy control plants. Keshmeshiye bidaneh cultivar showed the highest infection rate (16.74%), whereas Fakhri had the lowest infection level (9.83%).

### 3.2. Phylogenetic Analyses of the ORF5 Sequence

The partial ORF5 nucleotide sequences of eight Iranian isolates named T6, UC, VJ8, UN1, Cyl3, MR, Ben1, and NR were deposited in GenBank with accession numbers MG551301 to MG551308, respectively (Appendix A). Among them, T6, UC, and UN1 were from West Azarbaijan province, and VJ8, MR, Ben1, NR, and Cyl3 were from East Azarbaijan province. Sequence comparison showed that they were highly similar, with nucleotide identity scores ranging from 91.6 to 99.6%. The highest levels of nucleotide identity among them were observed for Ben1, Cyl3, and NR isolates, from East Azarbaijan province (99.6%), while MR and UN1 isolates shared 91.6% identity. Meanwhile, when we checked the identity of these new Iranian isolate sequences with three isolates previously reported from Southern Iran and other GVA sequences in the GenBank, scores of 90.8–99.2% and 87.4–95.4% were obtained, respectively. Considering the GVA sequences deposited in GenBank, the lowest identity (87.4%) occurred between the UN1 and 92PA isolates from Poland and the highest (95.4%) between the UC and TRAJ2-BR isolates from Brazil.

From the phylogenetic analysis of the partial ORF5 nucleotide sequences among the Iranian isolates determined in this study, the isolates previously reported from Iran and 45 isolates available in GenBank (Appendix A) revealed four major groups (Figure 1). There was no correlation between the geographical origin of these new Iranian isolates and their nucleotide sequences. All of the eight isolates analyzed in this study belong to group I; however, the isolates UN1, T6, Ben1, NR, Cyl3, MR, and UC clustered together, whereas VJ8 clustered with three previously reported Iranian GVA isolates. Group I also contained diverse isolates from different continents, whereas group II included mainly Chinese isolates, and groups III and IV included four divergent isolates of GVA.

### 3.3. Analysis of the Partial Protein Sequence of p10, the ORF5 Gene Product

The analysis of the partial p10 protein sequences of Iranian GVA isolates was conducted with Pfam, a widely used database of *protein* families and domains suitable to characterize a protein sequence. Based on Pfam, p10 was found to belong to the nucleic acid binding protein (NABP) family and the e-value was calculated as 3.2 × 10^−7^. The NABPs of the vitiviruses GVD, GVK and GVJ had the highest similarity with GVA p10, according to BLASTp results. The conserved sites of the RNA binding proteins have been determined for GVA and other vitiviruses including GVB, GVK, GVJ, GVD, GVH, AVA and AVB.

Multiple sequence alignment of these partial protein sequence (90 aa) using Weblogo showed significant amino acid conservation. The N-terminal portion of these proteins showed a basic, arginine-rich motif (ARM:KRRRARR), followed by a zinc-finger motif [C-X-C-X_4_ (GAIM)-H-X_4_ (NNKD)-C] (Figure 2A). As can be noted, the p10 protein of the GVA isolates here considered also includes a highly conserved domain HKLDRLRFVKEGRV of unknown function. Interestingly, while the arginine rich motif and the zinc finger motifs in the N-terminus of the protein are also conserved regions in the ORF5 products of other vitiviruses, including GVB, GVK, GVJ, GVD, GVH, AVA, and AVB (Figure 2B), the motif HKLDRLRFVKEGRV is not conserved (Figure 2B). Moreover, the C-terminal portion of the protein did not show any conserved region among these vitiviruses (Figure 2B).

### 3.4. Comparing the Topology of Full Genome Tree and Gene Trees

The nucleotide sequences of 18 complete genome sequenced GVA isolates were retrieved from NCBI GenBank (Appendix A). Since GVD is the most closely related virus out of the GVA groups based on BLAST results, the reference sequence of GVD (Acc. No. MF774336) was used as the out-group. Tree reconstruction based on individual genes and on full genome sequencing resulted in six trees. The best-fitting model selected by IQtree to reconstruct the full genome tree of the 18 GVA isolates based on BIC criteria was GTR + F + I + G4 [38]. The four-clade system of Alabi et al. [19] was used to name the main clades of the reconstructed tree (Figure 3a).

The GTR + F + I + G4 model was selected as the best-fitting one for ORF1, the TIM3 + F + G4 model for ORF2, the TIM2 + F + G4 model for ORF3, the K2P + G4 model for ORF4, the K2P + I + G4 model for ORF5, and the GTR +F + I + G4 model for the full genome [39,40]. The estimated total sites, informative parsimony characters, singletons, and constant sites, per reconstructed tree, as detected by IQtree, are summarized in Table 2. As indicated, ORF1 followed by ORF3 had the highest number of informative parsimony characters, whereas ORF5 had the least. Variable sites (informative characters and singleton sites) for all the five ORFs were as follows: ORF1, 64%; ORF2, 81%; ORF3, 61%; ORF4, 48%; and ORF5, 41%. As shown, ORF2 followed by ORF1 had the highest percentage of variable sites. In the next step, we compared the topology of the full genome and of the gene trees. For this, we first compared the order of clades as illustrated in Figure 3. The orders of the clades in individual gene trees were different from the full genome tree, except for ORF1. Clade IV in the ORF5 tree is merged with clade I. Additionally, the RF distance of the ORF1–ORF5 trees calculated by TreeCmp were 2, 5, 4, 8, and 8, respectively.

## 4. Discussion

In this study, GVA was detected in the main vineyards of East and West Azarbaijan provinces in Northwest Iran using both DAS-ELISA and RT-PCR with primers targeting ORF5. To date, 86 different viruses have been reported on grapevine worldwide and some of them have a strong economic impact on the cultivation of this crop [41]. The presence of viral diseases in commercial planting materials indicates the use of infected material for the establishment of vineyards and the lack of efficient monitoring systems for transferring and planting virus-free materials [42,43]. The management of viral disease of grapevine is very hard due to the lack of effective chemical compounds for their control, their fast evolution rate through mutation and genetic recombination events, their easy adaptation to the local environmental conditions, and the breakdown of plant genetic resistance [44]. For these reasons, it is very important to detect the virus/viruses infecting grapevine, especially from the perspective of ecologically sustainable agricultural practices and environmental preservation.

The most prevalent symptoms of GVA infection in the surveyed areas were leaf roll, leaf yellowing, leaf reddening, and irregular fruit ripening, as described previously in vineyards of Fars, East and West Azarbaijan, Kurdestan, *Kohgiluyeh* va *Boyer*-*Ahmad* and *Zanjan* provinces [10,13,14,15,16]. The plants analyzed in this study showed symptoms typical of GVA, but only 14% of them tested positive for this virus, implying that other viruses may be responsible for the symptoms.

Reconstruction of the phylogenetic tree using the partial ORF5 gene sequences of eight field isolates of GVA collected in the East and West Azarbaijan provinces discriminated four clades and all the newly sequenced Iranian isolates of this study clustered in group I. Our results are in agreement with previous studies that used different ORFs including CP, RdRp and partial nucleotide sequences encompassing the CP and ORF5 genes distinguishing GVA isolates into four groups [23,24,25,27].

In a second part of the work, the sequence of the ORF5 gene product, the p10 protein of the GVA Iranian isolates, was studied. The conserved sites of the p10 protein were determined and the Weblogo analysis showed the presence of a highly conserved basic, arginine-rich motif followed by a zinc-finger motif in its N-terminal portion. Our results are in agreement with previous studies showing that the basic, arginine-rich motif is responsible for the RNA binding activity of this protein [9,17]. Other vitiviruses used in this study showed conserved regions only in the N-terminal portion of the protein, with no conservation in the C-terminal part, emphasizing again the relevance of the N-terminal of the protein in RNA binding activity.

Different ORFs were used for phylogenetic analysis and evolutionary studies of GVA around the world. In this study, we compared the full genome tree and individual gene trees of GVA isolates using RF distance in order to find gene(s) suitable as evolutionary marker(s) in phylogenetic studies, instead of the whole viral genome. First, the topology and clade pattern of the full genome tree (as the true tree) were compared to those created with individual genes. All gene trees produced the same number of clusters as the full genome tree. A significant factor for selecting a phylogenetic marker is the level of sequence variability [45]; our results showed that ORF1 and ORF3 have the highest number of informative sites. Therefore, GTR +F + I + G4 was selected as the best model to reconstruct the tree for the full genome and ORF1. In summary, our results detected four main clades for all GVA isolates using either partial or complete sequences submitted to public databases. Therefore, ORF1 encoding RdRp can be an appropriate representative gene of the full genome in phylogenetic studies of GVA. It is worth indicating that a smaller, highly conserved part of this ORF may be sufficient, instead of the whole ORF. Indeed, the RdRp region of ORF1 was used by Alabi et al. (2014) to group the GVA isolates (19).

The selection of an appropriate phylogenetic gene marker was previously performed for other viruses by different authors. As an example, Tomimura et al. [46] indicated the NIb-VPg regions as phylogenetic markers for turnip mosaic virus, and Baradar et al. [47] reported that the NIb, Nib, and VPg genes can be used as evolutionary markers for bean yellow mosaic virus; both of these viruses belong to the *Potyvirus* genus. In this study, the ML-based IQtree and BI-based MrBayes programs were used for the reconstruction of full genome and individual gene trees and all fully sequenced GVA isolates were classified and compared to Alabi et al. [19]. In all the reconstructed trees, almost all main monophyletic groups detected by previous studies are present with high posterior probability and bootstrap support, named as groups I–IV, as suggested in the naming system by Alabi et al. [19]. The clades in all individual gene trees are almost similar to those of the full genome tree. Notably, when comparing individual genes and complete genome trees, our findings are in contrast with those of Alabi et al. [19] and with other studies that indicated that the GVA ORF4, encoding the CP, is the most suitable region for taxonomic purposes [10,20,21,22].

Most available isolates in public databases are partially sequenced and virologists are obliged to use partial genomes or individual genes for phylogenetic studies. As such, to avoid unbiased classification, phylogenetic studies should be based on either phylogenetic gene marker(s) or on a multigene dataset, instead of just one random gene (47). This study is the first to determine a marker gene for GVA phylogenetic and evolutionary analysis, and the methodology and the results here described can be used for further research on phylogenetic analysis, evolution history, epidemiology, etiology, and control measures of GVA, taken as the type species of the *Vitivirus* genus, but also for other viruses of grapevine. Moreover, protein sequence analyses of the partial ORF5 gene product of Iranian GVA isolates supports the RNA binding ability of this protein.

## Figures and Tables

**Figure 1 plants-11-01118-f001:**
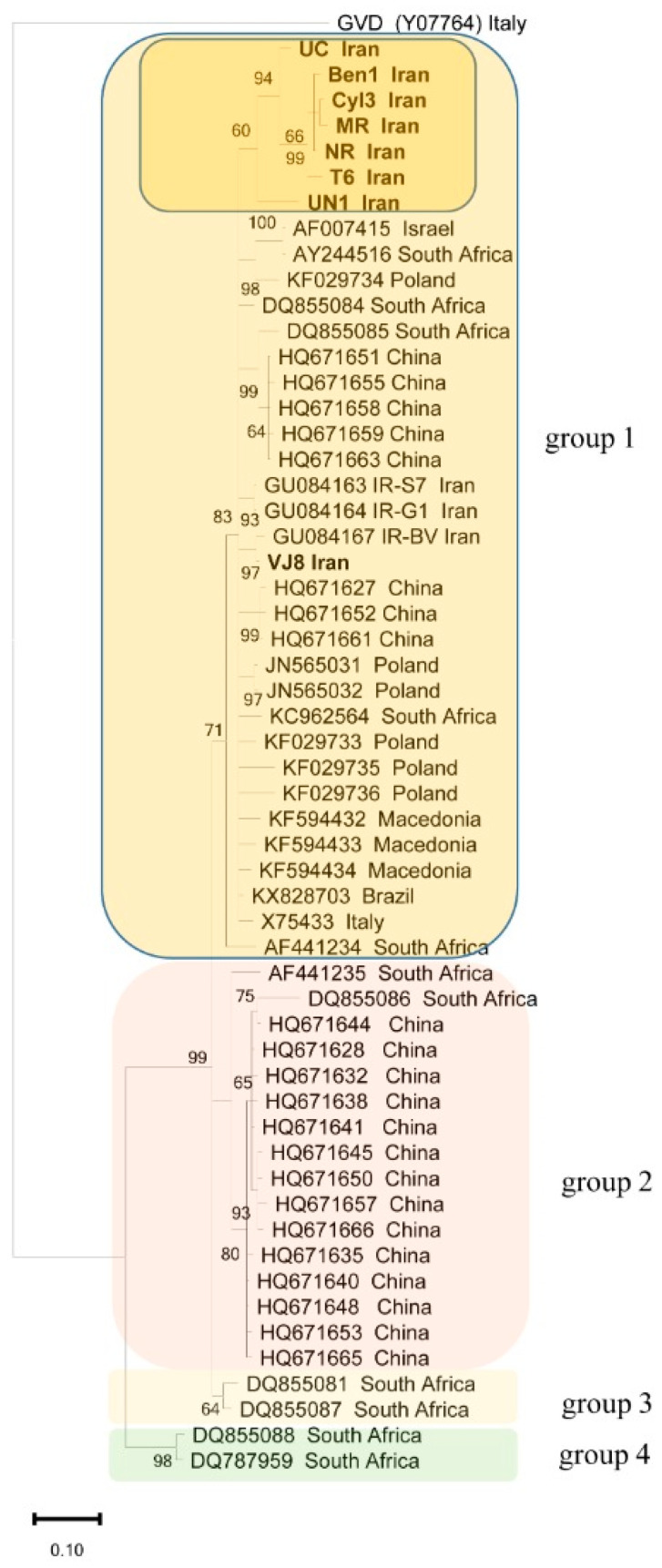
Phylogenetic tree of grapevine virus A (GVA) isolates based on partial nucleotide sequences of the ORF5 gene (236 nt) reconstructed by MrBayes. Bootstrap values (1000 replicates) are given at the branch nodes. Bootstrap values lower than 60 are not shown. Details of isolates are provided in Appendix A. Grapevine virus D (GVD) was used as the out-group. Iranian GVA isolates are shown in bold within a dark yellow box.

**Figure 2 plants-11-01118-f002:**
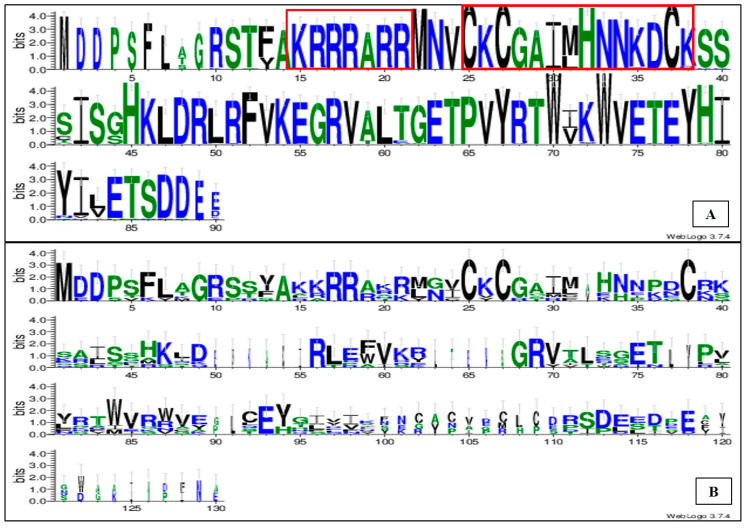
Weblogo depicting the multiple sequence alignment of the partial p10 protein sequences of the new GVA isolates collected in Iran (**A**). (**B**) shows the same analysis conducted including the RNA binding protein sequences of GVA, GVD (CAA69071), GVK (YP_009389467), GVJ (YP_009551971), GVH (YP_009551909), AVA (AET36889), and AVB (YP_004935362). The y-axis represents the bit score, where 4 means 100% conservation. The x-axis displays the amino acid position in the multiple sequence alignment. The arginine-rich motif (ARM) and the zinc-finger motif are shown in red rectangles.

**Figure 3 plants-11-01118-f003:**
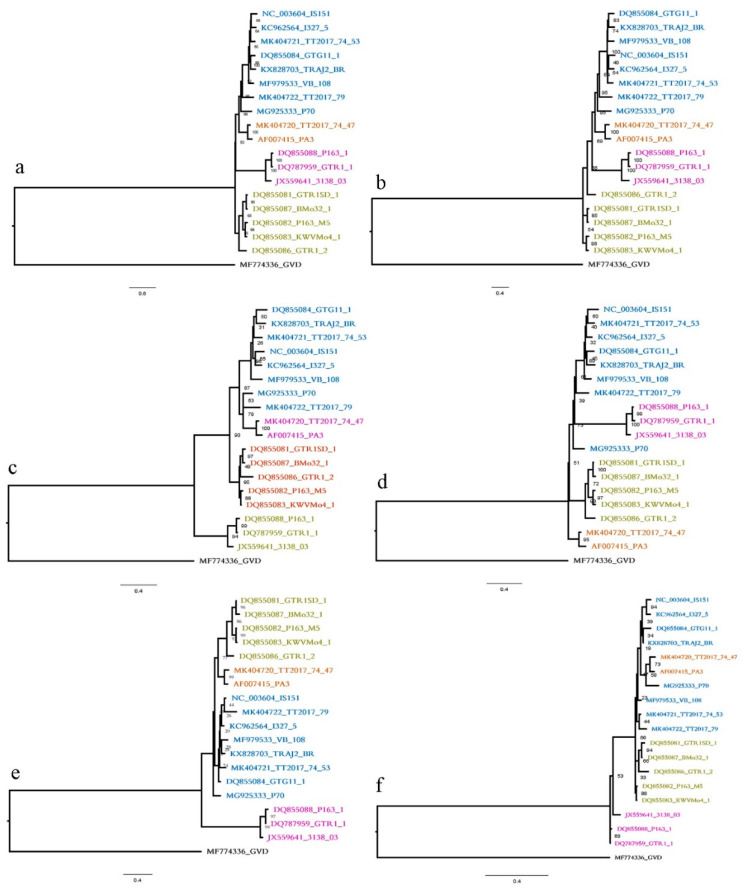
ML phylogenetic tree of (**a**) 18 terminals based on full genome sequences of all available GVA isolates and (**b**) 18 terminals based on ORF1, (**c**) ORF2, (**d**) ORF3, (**e**) ORF4, and (**f**) ORF5 sequences, all rooted by a GVD isolate. The classification is according to Alabi et al. [19]. Four clades, namely, I (blue), II (dark green), III (purple) and IV (brown) are shown.

**Table 1 plants-11-01118-t001:** Presence and distribution of GVA in grapevine samples in the East and West Azarbaijan provinces.

	East Azarbaijan Province	West Azarbaijan Province	Total
Cultivar	Infected/Tested (% Infection)	Infected/Tested (% Infection)	Infected/Tested (% Infection)
Keshmeshiye bidaneh	27/170 (15.9)	9/45 (20.0)	36/215 (16.7)
Angore-siyah	8/72 (11.1)	1/18 (5.6)	9/90 (10.0)
Fakhri	5/44 (11.4)	1/17 (5.9)	6/61 (9.8)
Garmian	3/23 13.0)	1/2 (50)	4/25 (16.0)
Qizil Uzum	1/8 (12.5)	0/4 (0)	1/9 (11.1)
Total	44/317 (13.9)	12/86 (14.4)	56/403 (14.0)

**Table 2 plants-11-01118-t002:** Characterization of the phylogenetic trees using IQtree.

	Best Model (Based on Bic)	Total Sites	Informative Sites	Singleton Sites	Constant Sites
Full genome tree	GTR + F+I + G4	7313	3408	1194	2711
ORF1 tree	GTR + F+I + G4	5126	2499	793	1834
ORF2 tree	TIM3 + F + G4	533	318	116	99
ORF3 tree	TIM2 + F + G4	837	374	137	326
ORF4 tree	K2P + G4	596	203	87	306
ORF5 tree	K2P + I + G4	273	58	54	161

## Data Availability

All the data is available in the manuscript.

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
