# Peer review of "Phylogenetic Marker Selection and Protein Sequence Analysis of the ORF5 Gene Product of Grapevine Virus A"

_plants, 2022, doi:10.3390/plants11091118_

Round 1
Reviewer 1 Report
The manuscript could be improved by paying attention to the following issues:
In the abstract, the authors described that GVA ORF1 was found to be the best phylogenetic marker. It would be nice to have a short description explaining the basis of the conclusion.
Details are missing in some parts of experiments. For example, information on the antibodies used in the DAS-Elisa experiment would be necessary and good for readers. It would also be necessary to describe the names of the specific isolates used in the multiple protein sequence analysis shown in Fig. 3.
Considering that the ORF1 comprises about 2/3 of the genome and sequence analysis of ORF1 takes as much effort as analyzing the whole genome, I wonder whether there is a big merit using ORF1 as a phylogenetic marker for GVA classification. If a smaller part of the ORF1 can be used as a marker, preferably if it is amplifiable by PCR, it would be great.
English in some parts of the manuscript, particularly in Discussion, looks strange. Thus, the manuscript needs to be read by a native English speaker.
Reviewer 2 Report
Manuscript ID plants-1668890 entitled "Phylogenetic marker selection and protein sequence analysis of the ORF5 gene product of Grapevine virus A" submitted by Mina Rastgou and coworkers provides additional information on the occurrence and partial sequences corresponding to ORF5 of GVA determined in eight isolates from East and West Azerbaijan provinces in the Northwest Iran. Comparison with GVA isolates determined in other viticultural regions of the world was performed using the partial ORF5 sequences. To find a suitable phylogenetic marker for GVA, the whole genome and five ORF sequences were compared separately, and ORF1 encoding RdRp was found to be the best marker for GVA classification. However, some parts of the manuscript are confusing and should be better explained. Samples for testing were initially selected based on symptoms such as yellowing/reddening and irregular ripening of berries. These symptoms were associated with GVA, but in contrast, the presence of GVA was confirmed in only 56 of 403 symptomatic samples (approximately 14%). This is clear evidence that the observed symptoms are not caused by GVA (or at least not only by GVA) but, on the contrary, are caused by other factors or are probably the result of mixed infections (GVA + viruses from the leafroll complex, which are very common in commercial vineyards). Testing the same samples for other economically important viruses (especially nepoviruses and viruses from the leafroll complex) might give a better insight into the actual causative agent(s) of the described symptoms. After partial ORF5 analyses and the construction of a phylogenetic tree, the authors speak of the existence of four groups and two subgroups. Such group separation is somewhat questionable when we consider the low branch support for these data (the authors mention a branch support of 60, perhaps lowering it to 50 would give better insight into clade/group separation). If lowering to 50% shows no difference (branches still without numbers), the entire phylogenetic tree is questionable or at least arbitrary. The authors should also clearly point out that not the whole ORF5 of the identified isolates is used for the analyses, but only a part of 236 nts (out of 272 nts). Another issue concerning the comparison of different ORFs is the number of informative parsimony sites, which should not be taken on its own but in comparison to the total number of nucleotides present in each ORF. For example, the authors point to ORF1 and ORF3 as the regions with the highest number of informative sites, but when compared to the length of ORF, ORF2 has the highest number of such sites (318/533 is about 59%, compared to ORF1 - 2499/5126 - about 49%, or ORF3 - 374/837 - about 45%). Also, the identification of ORF1 (RdRP) as the most suitable region of the whole genome for phylogenetic studies of GVA is not surprising, as it is generally a highly conserved region occupying about 70% of the whole genome. At the end of the discussion, the authors point out that this study is relevant for certification programmes for new vineyards and for management of virus diseases in existing vineyards - the authors are advised to describe in more detail the relationship between the above points.
There are also a lot of misleading references (30 in line 123, 21 in line 267, 40 in line 275, 45 in line 283, 23 in line 327, 21 in line 330).
Specific comments:
- According to the latest ICVT recommendations (and in the sense authors are using them ), virus names should not be written in italics, but in normal type (lines 16, 40, 211, 320, 321)
- Line 37 - since the number of grapevine viruses known today is well over 120, please be more specific or rephrase this sentence to something like According to available data from 2018[1], grapevine can be...
- Line 60 - frequently
- Lines 73 and 74 - is it necessary to write provinces in italics?
- Lines 90-92 - the number of samples analysed is a bit confusing - in the abstract and throughout the paper it is 403, here it is 425
- Lines 97-99 - clarify the information about the plant tissue collected and used for the laboratory analyses - unify this part
- Line 99 - specify the storage at 4°C and the time of the analyses performed, as the plant tissue degrades quickly at the temperature mentioned and could become unsuitable for analyses, especially molecular tests
- Line 108 - I assume that the authors mean the average value of negative controls, as at least two controls per test/plate are recommended
- Lines 119-123 - if the components used are part of a commercial kit, the authors should name them
- Lines 134-142 - here and throughout the manuscript, the authors should clearly indicate that only part of ORF5 was considered (236 nts out of a total of 272 nts)
- Line 148 - reference 32 should be normal and not bolded
- Line 162 - distances should be in normal style
- Figure 1 - since there is no clear relationship between the observed symptoms and GVA infection, the authors are advised to remove the photographs from the manuscript and not comment on the symptoms
- Line 187 - do you assume nucleotide identity?
- Line 191 - and other related sequences from other parts of the world...
- Line 192 - specify and mention that the comparison is done at the nucleotide level
- Line 196 - analysis of partial ORF5.
- Figure 2 - authors are advised to put the countries of origin in brackets after the acc. numbers + why were only 100 repeats used (not 1000) and why was a threshold of 60 set (not 50)?
- Line 224 - the word remarkable is too strong for this expression, as it is expected, the authors are advised to replace it with significant
- Figure 3 - the authors point out the conserved region in the N-terminus and select the KRRRARR and CKCGA... regions, but what about the section in the second line HKLDRLRFVKEGRV?
- Line 241 - 19 complete GVA sequences are mentioned in the text with reference to sup. t. 3, but only 18 sequences are listed in the table
- Figure 4 is fuzzy, authors should replace it with sharp one; also, according to the data presented and discussed in lines 327-328, there is some similarity between the full genome and ORF1 and 2, but not with others (ORF3-5) - authors should comment on this in more detail
- Line 274 - clarify with 86 GVA sequences, as over 600 GVA sequences are available in GenBank.
- Lines 267-277 - this sentence is a bit confusing, especially considering the possibility of vector transmission (mealybugs) documented for GVA
- Lines 289-290 - for what reason are the provinces italicised?
- References: consistent spelling of references - example 10 - one part is bold, 32nd part is underlined, etc.
- Supplementary Table 1 - authors are advised to describe the table in more detail and refer to the partial ORF5 used in the study + it would be useful to arrange the isolates in the following order: newly discovered, other already known from Iran, rest of the world, arranged by country of origin.
- Supplementary Table 2 - can be deleted and important data from it (alignment, HMM, bit score and e-value) can be inserted in the form of text below the results (lines 2019-220).
- Virus name (GVA) should be written normally and not in italics in all supplementary tables.
Reviewer 3 Report
I have added some brief comments and suggestions to the main document, but here are the some of the detailed points for areas that I feel were not clearly explained:
- The number of total plant samples used for analysis - there is contrasting information, 403 vs 425.
- For RT-PCR, what was the rationale for using/testing only the 56 that had tested positive by ELISA? What are the chances of getting false negatives by ELISA, and would these have contributed to the genetic diversity of the isolates in this study, possibly changing the phylogenetic results, for example?
- How many of the 56 ELISA-positive samples were positive by RT-PCR? Were they 8? If not, why were only 8 sequences used for further analysis? Were they representative enough of the geographic locations and/or cultivars used in the study?
- What was the input for the 8-sequence Pfam analysis? Were these the 8 sequences derived from PCR? If yes, I believe the PCR products represented partial ORF5. What % of the ORF was amplified/sequenced, and what difference would it make if the complete ORF sequence was used, instead?
- How many of the other GV species sequences were used for the amino acid alignment?
- What is the criteria used for labeling the 4 phylogenetic groups for the GVA sequences? Clustering or bootstrap support? You mentioned high posterior probability and bootstrap support, but I do not see significant bootstrap support for the major clades in Figs. 2 and 4. Also, in Fig. 4 the figure is not clear enough to see some values.
- If 403 symptomatic samples were collected, but only 56 turned out positive for GAV, what could be the reasons why this is so? I think this should have been included in the discussion, in addition to why only 8 sequences were used for analysis. I believe using more samples than these could have yielded different results.

Round 2
Reviewer 2 Report
Some points of the manuscript are better explained in the resubmitted version. For future work, the authors are advised to avoid ELISA on leaf samples collected in summer, as they can lead to a considerable number of false negatives - it is better to use the same tissue collected in autumn, or cortical scrapings from well-matured canes during dormancy. In Figure 1, where are the South Africa AF441234 and AF441235 isolates located? Do they form a separate group, or can they be included in groups 1 and 2? Although the quality of Fig. 3 has improved, it is still not sharp and clearly readable.
line 116 - controls
